# Double-Weighted Bayesian Model Combination for Metabolomics Data Description and Prediction

**DOI:** 10.3390/metabo15040214

**Published:** 2025-03-21

**Authors:** Jacopo Troisi, Martina Lombardi, Alessio Trotta, Vera Abenante, Andrea Ingenito, Nicole Palmieri, Sean M. Richards, Steven J. K. Symes, Pierpaolo Cavallo

**Affiliations:** 1Theoreo srl, Via degli Ulivi 3, 84090 Montecorvino Pugliano, SA, Italy; lombardi@theoreosrl.com (M.L.); trotta@theoreosrl.com (A.T.); abenante@theoreosrl.com (V.A.); ingenito@theoreosrl.com (A.I.); npalmieri@unisa.it (N.P.); 2European Institute of Metabolomics (EIM) Foundation, Via G. Puccini, 3, 84081 Baronissi, SA, Italy; 3Department of Medicine, Surgery and Dentistry “Scuola Medica Salernitana”, University of Salerno, 84081 Baronissi, SA, Italy; 4Department of Chemistry and Biology “A. Zambelli”, University of Salerno, 84084 Salerno, SA, Italy; 5Department of Biology, Geology and Environmental Sciences, University of Tennessee at Chattanooga, 615 McCallie Ave., Chattanooga, TN 37403, USA; sean-richards@utc.edu; 6Department of Obstetrics and Gynecology, Section on Maternal-Fetal Medicine, University of Tennessee College of Medicine, 979 East Third Street, Suite C-720, Chattanooga, TN 37403, USA; steven-symes@utc.edu; 7Department of Chemistry and Physics, University of Tennessee at Chattanooga, 615 McCallie Ave., Chattanooga, TN 37403, USA; 8Department of Physics, University of Salerno, 84084 Fisciano, SA, Italy; pcavallo@unisa.it; 9Istituto Sistemi Complessi—Consiglio Nazionale delle Ricerche, 00185 Rome, RM, Italy

**Keywords:** metabolomics, ensemble machine learning, Bayesian model, precision medicine, diagnostic tools

## Abstract

Background/Objectives: This study presents a novel double-weighted Bayesian Ensemble Machine Learning (DW-EML) model aimed at improving the classification and prediction of metabolomics data. This discipline, which involves the comprehensive analysis of metabolites in a biological system, provides valuable insights into complex biological processes and disease states. As metabolomics assumes an increasingly prominent role in the diagnosis of human diseases and in precision medicine, there is a pressing need for more robust artificial intelligence tools that can offer enhanced reliability and accuracy in medical applications. The proposed DW-EML model addresses this by integrating multiple classifiers within a double-weighted voting scheme, which assigns weights based on the cross-validation accuracy and classification confidence, ensuring a more reliable prediction framework. Methods: The model was applied to publicly available datasets derived from studies on critical illness in children, chronic typhoid carriage, and early detection of ovarian cancer. Results: The results demonstrate that the DW-EML approach outperformed methods traditionally used in metabolomics, such as the Partial Least Squares Discriminant Analysis in terms of accuracy and predictive power. Conclusions: The DW-EML model is a promising tool for metabolomic data analysis, offering enhanced robustness and reliability for diagnostic and prognostic applications and potentially contributing to the advancement of personalized and precision medicine.

## 1. Introduction

Over the last few decades, the field of “omics” technologies has brought about a transformative shift in the landscape of medical research, offering a revolutionary alternative to the traditional reductionist approaches that dominated biological investigation for much of the last century [1]. Traditional biology has often relied on the one-mutation–one-disease paradigm, wherein singular genetic mutations or alterations were studied in isolation to explain complex disease mechanisms. The omics sciences, with the advent of systems biology, have introduced a holistic approach to understanding the complexity of biological systems, emphasizing the interconnectedness of biological processes as a whole, rather than analyzing them in isolation [2,3].

While genomics allows us to predict potential biological outcomes based on an individual’s genetic pattern, metabolomics delves deeper into the current pathophysiological state by studying the entire set of metabolites, that is, the metabolome, within a biological system [4]. Comprising the substrates, intermediates, and products of the various biochemical reactions, the metabolites are the most sensitive indicators of the organism’s dynamic biochemical environment, reflecting shifts due to external stimuli and pathological conditions [5,6]. Hence, the aim of metabolomics investigations is not to identify an individual biomarker associated with the condition but to generate a comprehensive “signature” of metabolomic changes. This profiling approach not only offers valuable insights into the underlying mechanisms of various pathological phenotypes, enabling the generation of novel mechanistic hypotheses, but also has significant translational implications. Indeed, there is a growing application of metabolomics in the development of screening and diagnostic tests, as well as increasing interest in its potential applications for advancing personalized medicine approaches and precision medicine interventions [7].

Generally speaking, a metabolomic profiling assay allows the classification of subjects, for instance “normal” versus “pathologic”, by comparing the molecular profiles using machine learning tools to identify their biological or clinical status [8]. The process of metabolomic analysis is multifaceted, involving both experimental and computational components. The “wet” part of the analysis, which involves chemical assays, utilizes advanced techniques such as mass spectrometry (MS) or nuclear magnetic resonance (NMR) to detect and quantify metabolites. These methods enable high-throughput and highly sensitive detection of thousands of metabolites, generating extensive datasets. However, it is the “dry” part of the analysis—the computational and statistical processing of this vast amount of data—that often proves to be the most complex and time-consuming. Metabolomic data are typically structured as wide and short datasets, meaning that the number of variables (i.e., the metabolites) usually exceeds the number of samples.

While nearly all existing machine learning algorithms may be used in metabolomics, traditionally Partial Least Squares Discriminant Analysis (PLS-DA) has been the most widely utilized algorithm for metabolomic data analysis [9,10]. Its longevity can be attributed to both its practicality and its ability to simplify biological interpretation through the development of VIP (Variable Importance in Projection) scores. These scores highlight metabolites most relevant to class separation, facilitating a clearer understanding of the involved metabolic pathways and disease states. Consequently, as the PLS-DA’s utility has often overshadowed alternative approaches, other machine learning models have had limited adoption in the field [10]. However, as the field of metabolomics evolves, especially with increasing interest in its application for large-scale public health initiatives, there is a growing need for more robust and versatile classification models [11]. This need for improved performance is especially critical in the context of developing screening and diagnostic tests, where potential misclassifications, particularly false negatives, pose significant risks. In screening tests, a false negative—failing to identify a patient with the condition—can lead to delayed diagnoses, missed treatment opportunities, and even fatal outcomes. This is particularly dangerous in cases such as cancer screening, where early detection is crucial for improving survival rates, or in prenatal testing, where timely interventions can prevent severe complications [8].

To address this gap, we propose a Bayesian Ensemble Machine Learning (DW-EML) model that incorporates a double-weighting system. This approach is built on the principle of “ensembling” the results of multiple individual classification models rather than relying on a single algorithm to make predictions. The core idea is that this approach can mitigate the weaknesses of individual models in a collective decision-making process that makes the overall predictive performance more robust. Once each classifier has made its prediction for a sample, the DW-EML model introduces a voting scheme to combine these individual predictions into a final classification. The double-weighting is introduced in that each individual vote is weighted based on both the cross-validation accuracy and classification confidence (for instance, the distance from the classification margin) of that specific classifier. All of the double-weighted individual votes are then mathematically combined to result in a final ensembled prediction.

Our research group has already demonstrated the utility of this novel approach in various applications, including the development of oncology screening tests and prenatal diagnostics (see Appendix A). The aim of this study is to compare the performance of our DW-EML approach with that of conventional methods commonly used in metabolomics, applying our system to publicly available raw metabolomics datasets. Through this comparison, we seek to assess whether the DW-EML approach offers improvements in terms of accuracy, robustness, and predictive capability over traditional techniques, thus contributing to the advancement of diagnostic tools based on metabolomic analyses.

## 2. Materials and Methods

### 2.1. Data Collection

To assess the effectiveness of our proposed DW-EML as an alternative to traditional classification algorithms, we screened three publicly accessible databases of metabolomics data, including MetaboLights (https://www.ebi.ac.uk/metabolights/, access data: 22 May 2024), MetabolomeXchange (http://www.metabolomexchange.org/site/, access data: 22 May 2024), and the Metabolomics Workbench (https://www.metabolomicsworkbench.org/, access data: 22 May 2024), for relevant studies providing raw metabolomics data. The eligibility criteria for inclusion in this study required that the articles (a) focus on untargeted metabolomics, (b) adopt a case–case or case–control design that employed at least one classifier for discriminating between conditions, (c) reported the accuracy results of the trained model(s), and (d) publicly shared the datasets used for training the classifier. Three studies meeting these criteria were identified in the MetaboLights database, filtered by the organism *Homo sapiens*. The selected articles include: “NMR-based metabolic profiling provides diagnostic and prognostic information in critically ill children with suspected infection” [12], access number: MTBLS563; “Diagnostic metabolite biomarkers of chronic typhoid carriage” [13], access number: MTBLS579; and “Profiling the metabolome of uterine fluid for early detection of ovarian cancer” [14], access number: MTBLS4861.

From these studies, a total of eight datasets were extracted. These comprised three datasets from the first study (bacterial vs. control, viral vs. control, and bacterial vs. viral), four from the second study (Salmonella vs. control, S. Typhi vs. S. Paratyphi, S. Typhi vs. control, and S. Paratyphi vs. control), and one from the third study (ovarian cancer vs. ovarian benign training set). Each dataset was processed using our ensemble-based method, facilitating a direct comparison with the results obtained via traditional PLS-DA analysis.

### 2.2. Data Pretreatment

To ensure consistency between the data used to train the DW-EML model and those employed in the source articles, data normalization and pretreatment procedures were performed as specified in the original studies. When the reference articles did not provide pretreatment details, normalization by the total area was used as a default method. Specifically, the datasets from Grauslys et al. [12] were autoscaled, those from Näsström et al. [13] were normalized by the total area, and the dataset from Wang et al. [14] underwent normalization by the total area, followed by log_10_ transformation and Pareto scaling.

Moreover, to train our DW-EML model, a minimum of 100 total samples are required, so where necessary, synthetic samples were added using the SMOTE algorithm [15], keeping a check on possible class imbalance.

### 2.3. Ensemble Machine Learning

A total of nine machine learning algorithms were trained using Rapidminer 10.2 software, including Naïve Bayes (NB), Generalized Linear Model (GLM), Linear Regression (LR), Fast Large Margin (FLM), Deep Learning (DL), Decision Tree (DT), Random Forest (RF), Gradient Boosted Trees (GBT), and Support Vector Machine (SVM). Each classifier underwent cross-validation, and those achieving an accuracy ≥60% were selected to build the DW-EML model. The latter was generated using a double-weighted voting scheme. Specifically, each classifier’s prediction score was calculated and then multiplied by the cross-validation accuracy and classification confidence to generate a pre-score. The pre-score was then adjusted by a factor of +1 or −1, depending on whether the prediction was for “Case” or “Control”, respectively, resulting in that classifier voting score. The overall ensemble score was finally computed by summing the voting scores from all individual classifiers.

The ensemble score was then used with Matplotlib 3.8.2 (Python 3.11.9) to generate a Receiver–Operating Characteristic (ROC) curve, from which an optimal cut-off point was determined to maximize the Youden index (J) [16]. This cut-off value was subsequently used to create a confusion matrix for the ensemble model, allowing the calculation of overall classification accuracy.

For each dataset, the DW-EML model was trained under four different conditions: once with both hyperparameter optimization and feature selection, once with only hyperparameter optimization, once with only feature selection, and once without applying either. Moreover, all the trained DW-EML models underwent a permutation test, with *n* = 5 in order, to assess the possibility of over-fitting.

### 2.4. Hyperparameter Optimization and Feature Selection

Hyperparameter optimization was performed using a subprocessor from RapidMiner 10.2, comprising the following two key logical operators: one to set the hyperparameters and train each individual classifier, and another to identify the optimal combination of hyperparameter values. The optimization process employed the grid search technique, allowing us to evaluate various combinations and select the one that produced the best performance based on cross-validation results. As hyperparameters can vary greatly across classifiers, their optimization involved either choosing specific values or determining whether to apply a specific parameter. Specifically, in the case of NB, we optimized the Laplace correction. For the GLM, the parameters included family, solver, use of regularization, lambda, lambda search, alpha, standardization, non-negative coefficients, adding an intercept, removing collinear columns, maximum iterations, and specifying beta constraints. For the LR and FLM, we optimized classification strategies. In the case of the DL, parameters such as activation, hidden layer sizes, epochs, computation of variable importance, training samples per iteration, adaptive rate, epsilon, rho, standardization, L1 and L2 regularization, maximum W2, loss function, distribution function, and early stopping were optimized. For DT, we adjusted the criteria, maximum depth, pruning, pruning confidence, pre-pruning, minimum gain, minimum leaf size, minimum size for split, and number of pre-pruning alternatives.

For RF, the parameters included the number of trees, criteria, maximum depth, pruning, pre-pruning, random splits, subset guessing ratio, and voting strategy. GBT were optimized for the number of trees, maximum depth, minimum rows, minimum split improvement, number of bins, learning rate, sample rate, distribution, and early stopping. Finally, for SVM, we optimized SVM type, kernel type, γ, C, and ε. This approach ensured that each classifier was optimally fine-tuned through the selection of the most effective hyperparameters, improving overall performance across the models.

The feature selection process utilized a Genetic Algorithm (GA) in RapidMiner 10.2. This approach was configured with the following parameters: balance for accuracy set to 1.0 and local random seed set to 1992, with the optimization heuristic enabled. The process was constrained by a time limit of 400 s and a maximum function complexity of 3. The GA method explored various combinations of features, aiming to identify those that maximized the model’s performance.

## 3. Results

Here, we present the results obtained from the application of the DW-EML model to the datasets from the selected studies. Each dataset was analyzed under varying experimental conditions to assess the performance of the proposed method. We report the overall accuracy, area under the curve (AUC), and comparisons between our ensemble model and the results from the original articles.

The results are divided into three subsections, each focusing on one of the selected papers. Tables and visualizations are included to facilitate the comparison and highlight the improvements brought by our approach.

In most metabolomics studies, the data analysis phase often overlooks two critical aspects: hyperparameter optimization and feature selection. These steps, while frequently neglected, are essential components of our proposed ensemble method, as they contribute to the robustness of the model and help minimize the risk of overfitting. By carefully optimizing hyperparameters and selecting relevant features, our ensemble method can improve generalizability and enhance predictive accuracy, especially when dealing with complex, high-dimensional metabolomics data.

To ensure a thorough comparison with the methods reported in the original datasets, we present the results of both the fully optimized (FO) ensemble model—subjected to both feature selection and hyperparameter optimization—and simplified versions of the ensemble model without one or both of these steps. This allows us to highlight the added value of these optimization processes in improving model performance and increasing the comparability with the models reported in the original papers not subjected to these steps.

### 3.1. Study 1: NMR-Based Metabolic Profiling for Diagnostic and Prognostic Purposes in Critically Ill Children (Grauslys et al. [12])

Table 1 illustrates the performance of the DW-EML models under different model configurations, highlighting the overall accuracy obtained from our analysis of the datasets from the study by Grauslys et al. (2020) [12]. These datasets, referred to as CB, CV, and CB-CV, correspond to children with bacterial infections (CB) compared to controls, children with viral infections (CV) compared to controls, and children with bacterial infections compared to those with viral infections (CB-CV).

Each experimental configuration tested—such as those with and without feature selection (FS) or hyperparameter optimization (Opt)—required the training of a large number of models, ranging from 60,000 to 80,000, particularly for the configurations that included both FS and Opt. To ensure the robustness of the ensemble model and verify the absence of overfitting, each experimental condition was subjected to a permutation test, in which class labels were randomly assigned to the samples in the datasets. None of the models trained under these randomized conditions achieved an accuracy equal to or greater than 60%, which represents the threshold criterion for a model to be included in the construction of the ensemble.

These results strongly indicate that our ensemble model is not prone to overfitting, as it only retains classifiers that perform consistently above the established accuracy threshold when trained on non-randomized class labels. The results demonstrate a high level of accuracy in most scenarios, with some interesting variations based on whether feature selection or hyperparameter optimization was applied. For the CB dataset, the model achieved a peak accuracy of 95.2% when neither FS nor Opt was applied, slightly outperforming the setup in which both subprocesses were applied, which achieved 92.9%. When either FS or Opt was excluded, the accuracy decreased slightly, with values of 90.5% and 88.1%, respectively. Appendix A also reports several other metrics for the performance evaluation.

A similar trend was observed for the CV dataset, where the FO yielded an accuracy of 88.1%. When FS was omitted, the accuracy dropped to 81%, while the absence of Opt resulted in a decrease to 85.7%. Interestingly, when both FS and Opt were excluded, the model’s performance stabilized at 83.3%, suggesting that feature selection might play a more critical role in this case.

For the CB-CV dataset, the results were relatively consistent across the different experimental setups, with the highest accuracy (92.5%) observed both in the FO configuration and when neither feature selection nor optimization was applied. The removal of either FS and Opt alone did not substantially affect the performance, which stabilized at around 90%.

In addition to evaluating the overall classification accuracy, we also performed a detailed comparison of the AUC values using ROC curve analysis, which is displayed in Figure 1a,c,e. This figure illustrates the AUC performance of the four configurations of our ensemble model: the FO model with both FS and Opt, one model without FS, one without Opt, and one without either of these adjustments.

For the CB models, the FO configuration achieved an AUC of 0.966. When FS was omitted, the AUC dropped slightly to 0.936, and when trained without Opt, the AUC reached 0.945. Interestingly, when both FS and Opt were excluded, the AUC slightly improved to 0.973, underscoring the adaptability of the ensemble model in this scenario.

In the CV models, the FO configuration produced an AUC of 0.916. The exclusion of FS resulted in a more pronounced decrease, with the AUC dropping to 0.834, while the absence of Opt yielded an AUC of 0.889. When both processes were removed, the AUC stabilized at 0.875, suggesting that FS plays a more significant role in enhancing the model’s performance in this case.

For the CB-CV models, the FO configuration achieved the highest AUC, reaching 0.980. The removal of FS reduced the AUC slightly to 0.970, while removing Opt produced a similar AUC of 0.970. When both FS and Opt were excluded, the AUC remained robust at 0.977, indicating the overall resilience of the ensemble model even in less optimized configurations.

Furthermore, in Figure 1b,d,f, violin plots are shown for the FO ensemble model scores across the two classes for each dataset (CB, CV, and CB-CV). These plots illustrate the distribution of classification scores between the “Case” and “Control” groups, highlighting the quantitative differences and demonstrating the model’s discriminative power. The clear separation between the two classes, particularly in the CB-CV models, emphasizes the strong classification performance of our ensemble model.

Figure 2 presents histograms that directly compare the AUC values of our ensemble models, trained using the FO setup, with the models developed by Grauslys et al. [12]. These histograms provide a clear visual representation of the performance differences between the two approaches.

The first pair of histograms compares the AUC values for the CB dataset, where the original model developed by Grauslys et al. [12] achieved an AUC of 0.940, while our DW-EML outperformed it with an AUC of 0.966. In the second pair, focusing on the CV dataset, their model reached an AUC of 0.830, whereas our model achieved a significantly higher AUC of 0.916, demonstrating the effectiveness of our approach in this context. Finally, the third pair highlights the largest difference in the CB-CV dataset, where the original model obtained an AUC of 0.780, while the DW-EML model achieved a notably higher AUC of 0.980, indicating a substantial improvement in distinguishing between bacterial and viral infections.

We also conducted a direct comparison of the model performances on the CV dataset, assessing the impact of synthetic data generation via SMOTE versus no data augmentation as reported in Appendix A. With SMOTE applied, eight classifiers exceeded an accuracy threshold of 60%, and the ensemble model demonstrated an accuracy of 88.1%, sensitivity of 85 ± 8%, and specificity of 91 ± 6%. Conversely, without SMOTE, only six classifiers surpassed the same accuracy threshold, while the ensemble model achieved comparable performance, specifically an accuracy of 87.5%, sensitivity of 86 ± 9%, and specificity of 88 ± 6%. Although overall performance metrics were similar, the application of SMOTE increased the number of effective classifiers, thereby enhancing the overall robustness of the predictive system.

Table 2 displays the performance of the DW-EML models, trained with various optimizations, when applied to the four datasets of Näsström et al. [13], which include the ST, PT, ST-PT, and *Salmonella* datasets. The ST dataset compares samples with *Salmonella* Typhi to healthy controls, the PT dataset compares samples with *Salmonella* Paratyphi to healthy controls, the ST-PT dataset contrasts *Salmonella* Typhi with *Salmonella* Paratyphi samples, and the *Salmonella* dataset is designed to distinguish between samples affected with any type of *Salmonella* and non-carriers.

### 3.2. Study 2: Diagnostic Metabolite Biomarkers of Chronic Typhoid Carriage (Näsström et al. [13])

For the ST models, the FO configuration achieved a perfect accuracy of 100%. The accuracy remained at 100% without FS and slightly decreased to 97.5% when Opt was omitted. When both FS and Opt were excluded, the accuracy reached 97.6%, indicating the model’s robustness even without these optimizations.

In the PT models, the fully optimized configuration also yielded an accuracy of 100%, and this remained stable regardless of whether FS or Opt was applied. This suggests that the model was able to distinguish between Salmonella Paratyphi and controls with high reliability under all configurations.

For the ST-PT models, the accuracy was consistently 100% across all experimental setups, with no significant drop in performance, regardless of whether FS or Opt was applied. This further reinforces the high discriminative power of the DW-EML in distinguishing between different Salmonella strains.

Finally, for the Salmonella models, the FO setup achieved an accuracy of 100%. However, when FS was omitted, the accuracy dropped slightly to 97.5%, while removing hyperparameter optimization maintained an accuracy of 100%. When FS and Opt were excluded, accuracy remained at 97.5%.

In Figure 3, the AUC values for each version of our models using ROC curve analysis are reported. This includes models with both FS and Opt, models without FS, models without Opt, and models without either. For the ST models, the FO configuration achieved an AUC of 1, and this remained unchanged when FS was excluded. The AUC dropped slightly to 0.998 without Opt and decreased further to 0.995 when both FS and Opt were omitted.

For the PT models, the AUC remained at a perfect score of 1 across all configurations, highlighting the model’s exceptional performance in this dataset.

In the ST-PT models, the lowest AUC was observed with the FO setup at 0.998, while the AUC improved to 1 in all other configurations, demonstrating that FS and Opt may not be as critical in this case.

For the Salmonella models, the FO configuration achieved an AUC of 1. When FS was excluded, the AUC dropped slightly to 0.997, while without Opt, it remained at 1. When both FS and Opt were removed, the AUC reached 0.998, confirming the overall robustness of the ensemble model.

Finally, in Figure 4, we display histograms comparing the AUC values of DW-EML models trained using the FO setup with those reported by Nasstrom et al. (2018) [13]. The first pair of histograms contrasts their ST model, which achieved an AUC of 0.950, with the DW-EML model, which reached a perfect AUC of 1. The second pair compares their PT model, with an AUC of 0.990, against the DW-EML, which also achieved an AUC of 1. The third pair illustrates the ST-PT model, for which their original model had an AUC of 0.833, while the DW-EML performed significantly better with an AUC of 0.998. The final pair compares their Salmonella model, which had an AUC of 0.974, to the DW-EML model’s AUC of 1, further showcasing the performance improvements offered by our ensemble approach.

### 3.3. Study 3: Profiling the Metabolome of Uterine Fluid for Early Detection of Ovarian Cancer (Pan Wang et al. [14])

Similarly to the other two studies, Table 3 presents the performance of the DW-EML across the various experimental conditions for the ovarian cancer (OC) dataset, which compares samples from ovarian cancer patients to controls.

For the OC model, the FO setup achieved an accuracy of 95%. The accuracy remained stable at 95% even when FS was excluded but dropped slightly to 92.5% when Opt was omitted. Interestingly, when FS and Opt were excluded, the accuracy rose again to 95%, indicating the ensemble model’s robustness and adaptability in this particular dataset.

In Figure 5, the ROC curve analysis, which compares the AUC values across the four versions of the DW-EML ensemble model were reported. The FO configuration achieved an AUC of 0.990. When FS was excluded, the AUC dropped to 0.980, and the same value of 0.980 was observed when Opt was not applied. In contrast, the AUC slightly improved to 0.985 when FS and Opt were omitted, reinforcing the ensemble model’s strong performance even with fewer optimizations.

Finally, Figure 6 presents a histogram comparing the AUC values of the DW-EML model with that developed by Wang et al. [14]. This histogram visually highlights the performance differences between their original OC model and DW-EML model, both trained on the same dataset. Specifically, the AUC of their OC model was reported as 0.940, while the DW-EML model outperformed it, achieving an AUC of 0.990, demonstrating a clear improvement in classification performance.

## 4. Discussion

This study aimed to evaluate the effectiveness of the DW-EML model for the classification and prediction of metabolomics data. The proposed model integrates multiple classifiers with a double-weighted voting scheme, which assigns weights based on cross-validation accuracy and classification confidence. This approach was applied to predict several pathological conditions both oncological and non-oncological (see Appendix A). Here we applied it to publicly available datasets to compare its performance with traditional methods, such as partial least squares discriminant analysis. The results demonstrate that the DW-EML model outperformed traditional approaches in terms of accuracy and predictive capability, thus offering an enhanced tool for metabolomic analysis and highlighting its potential to contribute to the development of robust screening and diagnostic tools. Moreover, unlike previous studies where we applied this framework to datasets collected under uniform conditions within our laboratory, this work serves as an independent validation of the approach on externally generated data. The fact that our model maintains high performance despite variations in sample collection and analytical methodologies underscores its flexibility and potential applicability in real-world scenarios.

This aspect is particularly relevant in biomedical research, where data acquisition protocols often differ across institutions and studies. The ability of our ensemble method to generalize across these variations suggests that it can be reliably applied in heterogeneous settings without requiring extensive recalibration. This is a crucial advantage over traditional models that tend to suffer from domain shifts when applied to datasets obtained under different conditions. Robust classification systems are crucial in metabolomics, especially given the potential applications of these tools in secondary prevention of human diseases and precision medicine. Metabolomics is increasingly used to identify disease signatures that could inform patient management decisions, predict health outcomes, and assist in the early detection of diseases [17]. As these techniques move closer to clinical applications, the robustness and reliability of classification models become paramount [18]. The ability to use models such as the DW-EML for creating diagnostic or screening tests could significantly advance personalized medicine, where early intervention can drastically improve patient outcomes. This is particularly critical in applications like oncology screenings or prenatal diagnostics, where early and precise identification of pathological states is key to successful intervention.

The proposed DW-EML showed very good classification results. One of the key reasons in its intrinsic design. Specifically, for a misclassification to occur, more than half of the classifiers within the ensemble must produce the same incorrect prediction. This is a crucial property, as it significantly reduces the likelihood of systematic errors affecting the final classification. Unlike traditional ensemble methods that aggregate models with similar training logics, our approach leverages a diverse set of classifiers, each trained using distinct methodologies and decision rules. This heterogeneity enhances robustness by minimizing the risk of correlated errors across classifiers. When models rely on different learning paradigms, it becomes increasingly improbable that multiple classifiers will make the same mistake on a given sample. Instead, their varying perspectives contribute to a more balanced and reliable decision-making process. This mechanism explains why the ensemble consistently achieves strong generalization performance, even in the absence of external validation datasets. Moreover, the weighting of individual model credibility and classification confidence further strengthens its robustness, ensuring that the overall decision-making process remains stable and reliable across different scenarios.

Additionally, both the individual scores and the overall score provided by the DW-EML can serve as specific evaluative elements of the classification capabilities of the models, thereby helping clinicians assess the appropriateness of the obtained response. This, combined with insights into the most relevant metabolites used by each model for classification, meets the need for interpretable models whose predictions can also be evaluated by human experts [19]. Indeed, the DW-EML approach also provides valuable insights into the relative importance of individual metabolites in the training of each classifier. By evaluating which metabolites carry the most weight in each model, researchers can develop a combined perspective that broadens the range of hypotheses that can be generated from untargeted metabolomic data using only the PLS-DA. Metabolites selected consistently across different classifiers are likely to have significant biological relevance, and their consistent identification across multiple models strengthens confidence in their role. This level of assessment is only possible when multiple classifiers are combined, as the DW-EML approach does, making it a powerful tool for understanding complex metabolomic data and generating new biological insights.

Furthermore, the DW-EML provides a quantitative score that serves as a basis for prediction, allowing for the flexibility to adjust the classification threshold to suit different clinical needs. This flexibility is particularly advantageous in settings where minimizing specific classification errors is crucial. For example, in oncology screenings, false negatives are more problematic than false positives, as missing a diagnosis can lead to delayed treatment and worse outcomes. Conversely, false positives, though inconvenient, can be addressed through follow-up diagnostic tests. By adjusting the cut-off score, the DW-EML can be tailored to minimize the risk of false negatives, optimizing its use for different medical applications where the stakes of misclassification differ. Such adaptability is not possible with models that provide only a binary healthy/diseased classification, making the DW-EML a superior option for scenarios requiring nuanced decision-making.

This work aligns well with existing literature on ensemble learning approaches in metabolomics and bioinformatics. Previous studies have demonstrated that ensemble deep neural networks (EDNNs) outperform traditional deep neural networks, random forest, and support vector machine algorithms in metabolomics regression tasks, showing significant improvements in predictive accuracy [20]. Similarly, ensemble learning models, such as stacking, bagging, and boosting, have been noted for their superior performance in disease prediction tasks, which can also be extended to metabolomics [21,22]. The DW-EML model builds on this concept by incorporating a double-weighting mechanism, further enhancing robustness and accuracy in metabolomic data classification.

Ensemble feature selection methods are also known to improve the reliability and stability of biomarker discovery by combining multiple feature selection algorithms, which is crucial given the high dimensionality and small sample sizes typical of metabolomics datasets [23,24]. This approach outperforms individual feature selection techniques, providing more stable and accurate identification of discriminative features [25]. The DW-EML model benefits from this ensemble approach by identifying key metabolites across a variety of classifiers, adding a layer of reliability to biomarker discovery.

Moreover, ensemble methods have proven effective in handling technical variations in metabolomics data. For instance, the TIGER method, an ensemble learning architecture, has been utilized to eliminate technical variations in large datasets, thereby improving the robustness of data analysis [25]. Similar benefits can be inferred from the DW-EML model, which also integrates multiple classifiers to mitigate variability and improve reliability.

Ensemble-based approaches have also been applied successfully to classify metabolite time series data, achieving significant performance even in challenging scenarios where group differences are subtle [26]. The versatility of ensemble learning methods, including their use in gene expression analysis, proteomics, and regulatory element prediction, demonstrates their broad applicability and effectiveness in handling complex biological data [27,28]. The DW-EML model is consistent with these findings, offering a powerful and adaptable tool for diverse applications in bioinformatics.

This study has several limitations. Firstly, relying on previously published data might be considered a weakness in substantiating the widespread use of the DW-EML in metabolomics. However, as detailed in Appendix A, several studies utilizing original datasets have been published over the past decade, with two of these also including findings from independent prospective cohorts—thereby reinforcing the robustness of the DW-EML approach. Another limitation concerns the use of the SMOTE algorithm for addressing class imbalance. Although SMOTE is necessary to mitigate overfitting and avoid a training bias toward the most represented class, it may introduce bias or distort the true distribution of metabolomic data. Moreover, the relatively small number of samples compared to the high dimensionality of the features makes under-sampling techniques less conventional in metabolomics, thereby supporting the choice to employ SMOTE algorithms. Furthermore, although the primary objective of this paper is to propose the DW-EML as a robust alternative to the prevalent use of the PLS-DA in metabolomics, the combination of machine learning algorithms employed was not optimized. The reported results could benefit from a more targeted selection of ML models, as well as a different strategy of feature selection, potentially leading to further performance improvements. Overall, the findings of this study suggest that the DW-EML model offers an effective, flexible, and insightful approach to metabolomic classification. By combining multiple classifiers, optimizing feature selection, and providing a scoring system that can be tuned to clinical needs, the DW-EML advances the potential for metabolomics to contribute meaningfully to precision medicine and early diagnostic strategies. Future studies should explore its application to larger datasets and additional clinical scenarios to further validate and expand its utility.

## Figures and Tables

**Figure 1 metabolites-15-00214-f001:**
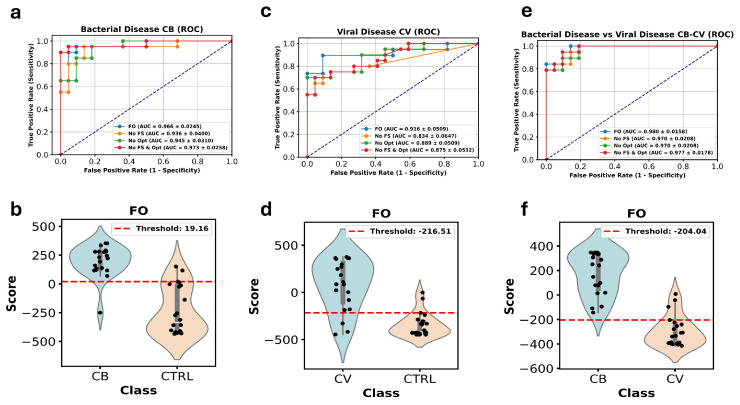
Performance comparison of the DW-EML across different experimental configurations for the CB, CV, and CB-CV datasets. (**a**,**c**,**e**) ROC curves displaying the AUC values for four model configurations: the complete model with both feature selection and hyperparameter optimization (blue line), the model without feature selection (yellow line), the model without hyperparameter optimization (green line), and the model without either feature selection or optimization (red line). (**b**,**d**,**f**) Violin plots of the FO ensemble model scores for the “Case” and “Control” groups in the CB, CV, and CB-CV datasets, respectively. Red dotted line represents the threshold value according to Youden index.

**Figure 2 metabolites-15-00214-f002:**
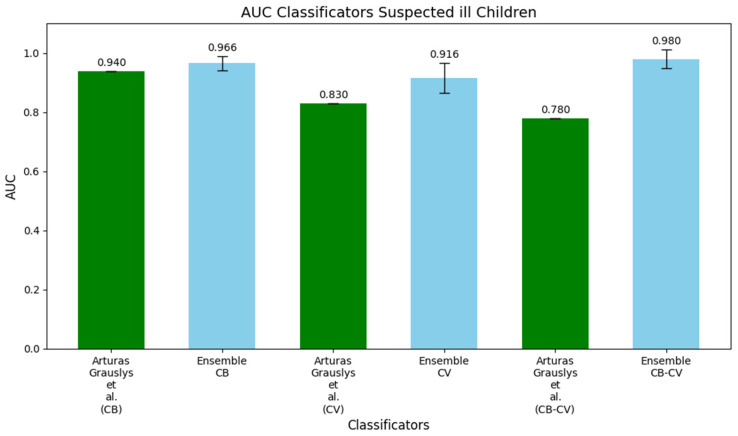
Histograms comparing the AUC values of the fully optimized ensemble machine learning models with those reported by Grauslys et al. (2020) [12] for the CB, CV, and CB-CV datasets.

**Figure 3 metabolites-15-00214-f003:**
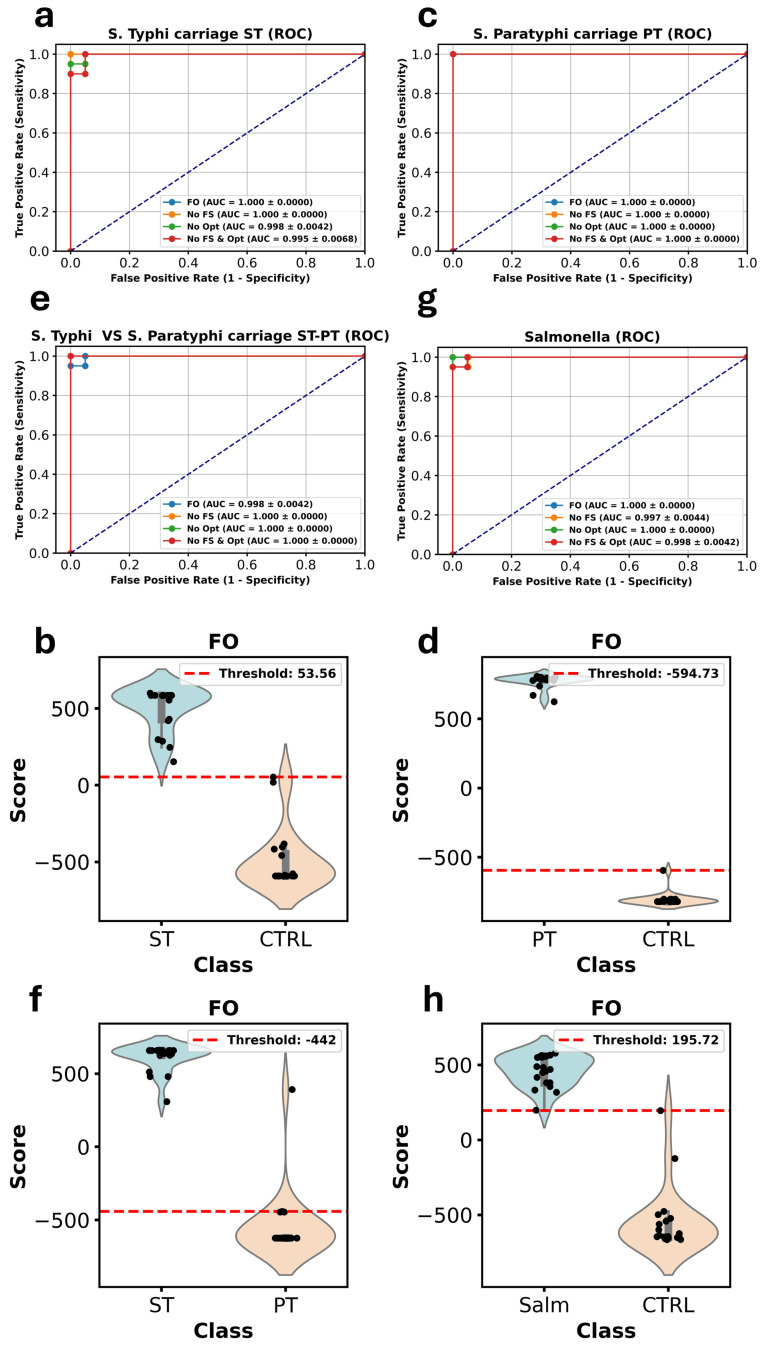
ROC curve analysis showing the AUC values of the ensemble machine learning models across different experimental settings for the ST, PT, ST-PT, and Salmonella datasets (**a**,**c**,**e**,**g**). ROC curves displaying the AUC values for four model configurations: complete model with both feature selection and hyperparameter optimization (blue line), the model without feature selection (yellow line), the model without hyperparameter optimization (green line), and the model without either feature selection or optimization (red line). The plots show the consistent high performance of the ensemble models in distinguishing between cases and controls across all datasets (**b**,**d**,**f**,**h**).

**Figure 4 metabolites-15-00214-f004:**
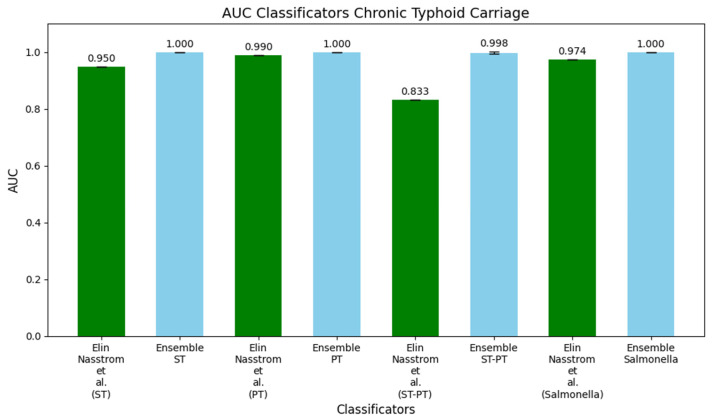
Histograms of AUC values of the ensemble machine learning models trained with full optimization compared with the results reported by Näsström et al. [13] for the ST, PT, ST-PT, and Salmonella datasets. The performance of the DW-EML model consistently yielded higher AUC values compared to the original analyses.

**Figure 5 metabolites-15-00214-f005:**
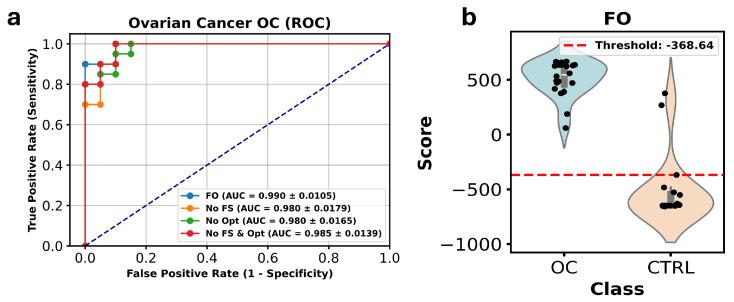
ROC curve analysis of the ensemble machine learning models across different experimental configurations for the OC dataset. ROC curves displaying the AUC values for four model configurations: the complete model with both feature selection and hyperparameter optimization (blue line), the model without feature selection (yellow line), the model without hyperparameter optimization (green line), and the model without either feature selection or optimization (red line). The AUC values demonstrate the model’s robust performance across all configurations, with the fully optimized model achieving the highest AUC of 0.990 (**a**). The plot highlight the reliable effectiveness of the ensemble model in differentiating between cases and controls (**b**).

**Figure 6 metabolites-15-00214-f006:**
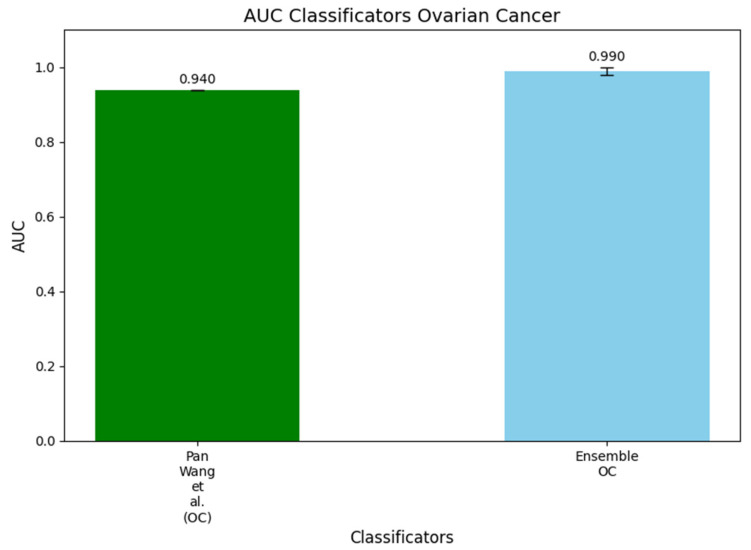
Histogram comparing the AUC values of the ensemble machine learning model trained using the classic setup with the AUC reported by Wang et al. [14] for the OC dataset. Their model achieved an AUC of 0.940, while our ensemble model, incorporating feature selection and hyperparameter optimization, significantly outperformed it with an AUC of 0.990, highlighting the improvements in predictive performance offered by the ensemble approach.

**Table 1 metabolites-15-00214-t001:** Classification accuracy of ensemble machine learning models applied to dataset reported in [12] across different experimental configurations and compared to individual classifiers. DT = Decision Tree; FLM = Fast Large Margin; FO = Full Optimization (both FS and Opt); FS = Features Selection; GBT = Gradient Boost Tree; GLM = Generalized Linear Model; LR = Logistic Regression; NB = Naïve Bayes; Opt = Hyperparameters Optimization; RF = Random Forest; SVM = Support Vector Machine. “-” indicates that model showed performance < 60%.

	Configuration	Ensemble	NB	GLM	LR	FLM	DL	DT	RF	GBT	SVM
Accuracy CB Model	FO	92.9%	-	96.7%	-	73.3%	83.3%	70.0%	60.0%	86.7%	-
	No FS	90.5%	-	93.3%	60.0%	80.0%	80.0%	-	63.3%	60.0%	80.0%
	No Opt	88.1%	-	83.3%	-	73.3%	83.3%	-	-	76.7%	73.3%
	No FS and Opt	95.2%	-	93.3%	60.0%	70.0%	80.0%	63.3%	90.0%	73.3%	90.0%
Accuracy CV Model	FO	88.1%	73.3%	86.7%	-	86.7%	76.7%	70.0%	63.3%	60.0%	73.3%
	No FS	81%	70%	73.3%	-	73.3%	86.7%	70.0%	73.3%	63.3%	73.3%
	No Opt	85.7%	73.3%	73.3%	-	66.7%	76.7%	73.3%	80.0%	70.0%	-
	No FS and Opt	83.3%	70%	73.3%	-	76.7%	86.7%	73.3%	70.0%	63.3%	80.0%
Accuracy CB-CV Model	FO	92.5%	86.7%	78.7%	-	96.7%	79.3%	72.7%	72.7%	78.7%	86.0%
	No FS	90%	68%	93.3%	-	90.0%	90.0%	83.3%	68.7%	78.7%	82.7%
	No Opt	90%	86.7%	86.7%	-	90.0%	83.3%	-	-	79.3%	79.3%
	No FS and Opt	92.5%	68%	93.3%	-	86.7%	90.0%	76.7%	75.3%	78.7%	93.3%

**Table 2 metabolites-15-00214-t002:** Performance of ensemble machine learning models applied on dataset reported in [13] across different experimental settings. DT = Decision Tree; FLM = Fast Large Margin; FO = Full Optimization (both FS and Opt); FS = Features Selection; GBT = Gradient Boost Tree; GLM = Generalized Linear Model; LR = Logistic Regression; NB = Naïve Bayes; Opt = Hyperparameters Optimization; RF = Random Forest; SVM = Support Vector Machine. “-” indicates that model showed performance lower than 60%.

	Configuration	Ensemble	NB	GLM	LR	FLM	DL	DT	RF	GBT	SVM
Accuracy ST Model	FO	100.0%	90.0%	96.7%	-	93.3%	89.3%	90.0%	100.0%	93.3%	100.0%
	No FS	100.0%	100.0%	100.0%	96.7%	100.0%	96.7%	93.3%	96.7%	90.0%	100.0%
	No Opt	97.5%	90.0%	96.7%	89.3%	-	100.0%	93.3%	93.3%	90.0%	-
	No FS and Opt	97.6%	100.0%	100.0%	96.7%	90.0%	96.7%	93.3%	93.3%	90.0%	-
Accuracy PT Model	FO	92.9%	-	96.7%	-	73.3%	83.3%	70.0%	60.0%	86.7%	-
	No FS	90.5%	-	93.3%	60.0%	80.0%	80.0%	-	63.3%	60.0%	80.0%
	No Opt	88.1%	-	83.3%	-	73.3%	83.3%	-	-	76.7%	73.3%
	No FS and Opt	95.2%	-	93.3%	60.0%	70.0%	80.0%	63.3%	90.0%	73.3%	90.0%
Accuracy ST-PT Model	FO	88.1%	73.3%	86.7%	-	86.7%	76.7%	70.0%	63.3%	60.0%	73.3%
	No FS	81%	70%	73.3%	-	73.3%	86.7%	70.0%	73.3%	63.3%	73.3%
	No Opt	85.7%	73.3%	73.3%	-	66.7%	76.7%	73.3%	80.0%	70.0%	-
	No FS and Opt	83.3%	70%	73.3%	-	76.7%	86.7%	73.3%	70.0%	63.3%	80.0%
Accuracy Salmonella Model	FO	92.5%	86.7%	78.7%	-	96.7%	79.3%	72.7%	72.7%	78.7%	86.0%
	No FS	90%	68%	93.3%	-	90.0%	90.0%	83.3%	68.7%	78.7%	82.7%
	No Opt	90%	86.7%	86.7%	-	90.0%	83.3%	-	-	79.3%	79.3%
	No FS and Opt	92.5%	68%	93.3%	-	86.7%	90.0%	76.7%	75.3%	78.7%	93.3%

**Table 3 metabolites-15-00214-t003:** Performance of ensemble machine learning models applied on dataset reported in [14] across different experimental settings. DT = Decision Tree; FLM = Fast Large Margin; FO = Full Optimizing (both FS and Opt); FS = Features Selection; GBT = Gradient Boost Tree; GLM = Generalized Linear Model; LR = Logistic Regression; NB = Naïve Bayes; Opt = Hyperparameters Optimization; RF = Random Forest; SVM = Support Vector Machine. “-” indicates that model showed performance lower than 60%.

	Configuration	Ensemble	NB	GLM	LR	FLM	DL	DT	RF	GBT	SVM
Accuracy OC Model	FO	95.0%	93.3%	96.0%	-	90.0%	88.7%	72.7%	96.7%	96.7%	96.7%
	No FS	95.0%	75.3%	93.3%	90.0%	89.3%	96.7%	81.3%	90.0%	85.3%	96.7%
	No Opt	92.5%	75.3%	96.0%	-	86.0%	92.7%	88.7%	86.0%	68.7%	92.7%
	No FS & Opt	95.0%	75.3%	93.3%	90.0%	89.3%	96.7%	84.7%	96.7%	72.0%	96.7%

## Data Availability

No new data were created or analyzed in this study. Data are contained within the article and Appendix A.

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
