# Peer review of "Double-Weighted Bayesian Model Combination for Metabolomics Data Description and Prediction"

_metabolites, 2025, doi:10.3390/metabo15040214_

Round 1

Reviewer 1 Report

Comments and Suggestions for Authors

The manuscript in question contains some typos and grammatical errors. Additionally, here are my concerns that addressing them can make this manuscript suitable for publication.

1- There are many references missing. I provided some urgent citations for some sentences, but not enough for the whole manuscript. For example, lines 55-58 (https://doi.org/10.1371/journal.pone.0282316). 

2- The study employs a variety of machine learning classifiers, but the rationale for selecting these specific models is not fully explained. Were other state-of-the-art classifiers (e.g., XGBoost or deep learning variations) considered?

3- The use of SMOTE to generate synthetic samples raises concerns about whether this could introduce bias or distort the real distribution of metabolomic data.

4- While a Genetic Algorithm (GA) is used for feature selection, it is unclear whether alternative feature selection methods were tested to ensure robustness.

5- The authors rely solely on publicly available datasets. The generalizability of the proposed model would be more rigorously assessed if validated on an independent, unseen dataset.

6- Despite the claim that permutation tests confirmed robustness, the very high accuracy scores (especially near 100%) suggest possible overfitting, particularly since metabolomic datasets are often noisy and variable.

7- The authors use multiple datasets from different sources but do not discuss potential batch effects, variations in sample collection, or differences in analytical methods.

8- The authors compare the proposed model with traditional PLS-DA but do not benchmark against other widely used ensemble learning techniques.

9- It is unclear whether statistical tests (e.g., paired t-tests or Wilcoxon signed-rank tests) were conducted to confirm that the improvements are significant.

10- The DW-EML model remains unclear on how metabolite importance is interpreted. Can the model outputs be linked to meaningful biological insights?

11- The paper does not specify whether the code, scripts, and parameter settings used for model training are available for reproducibility.

Author Response

The manuscript in question contains some typos and grammatical errors. Additionally, here are my concerns that addressing them can make this manuscript suitable for publication.

  • There are many references missing. I provided some urgent citations for some sentences, but not enough for the whole manuscript. For example, lines 55-58 (https://doi.org/10.1371/journal.pone.0282316). 

Reply: In response to the suggestion, we have included the paper on the network analysis of transcriptomics and metabolomics in Rosmarinus officinalis. However, we acknowledge that its relevance to the overall manuscript topic and to the specific sentence is limited.

  • The study employs a variety of machine learning classifiers, but the rationale for selecting these specific models is not fully explained. Were other state-of-the-art classifiers (e.g., XGBoost or deep learning variations) considered?

Reply: The DW-EML represents an ensemble framework capable of integrating various classification models. In this study, the proposed ensemble incorporates multiple machine learning techniques, including decision tree–based methods (though not specifically the XGBoost model) and a deep learning model. With the abundance of available ML models, this framework is inherently flexible, allowing for the potential inclusion of many alternatives. Importantly, the objective of this paper is not to identify the optimal combination for a specific scenario, but rather to demonstrate the relative advantage of using a single model for analyzing metabolomics data. We also included this in the study limitation.

  • The use of SMOTE to generate synthetic samples raises concerns about whether this could introduce bias or distort the real distribution of metabolomic data.

Reply: We acknowledge this consideration as a study limitation. Nevertheless, it is important to emphasize that managing class imbalance is essential when training a machine learning system. In metabolomics, the inherent scarcity of samples relative to the number of features renders undersampling strategies impractical, thus necessitating alternative approaches.

  • While a Genetic Algorithm (GA) is used for feature selection, it is unclear whether alternative feature selection methods were tested to ensure robustness.

Reply: The genetic algorithm was chosen for feature selection because it is the one that best fits the Bayesian ensemble scheme we proposed in this study. In fact, by its very nature, this algorithm selects the features that best train any classifier using accuracy as fitness. Alternative feature selection tools are more prone to introduce bias or fit only some specific classifiers. see in this regard (10.1016/B978-0-323-85062-9.00009-X).

  • The authors rely solely on publicly available datasets. The generalizability of the proposed model would be more rigorously assessed if validated on an independent, unseen dataset.

Reply: In lines 110–118 of the introduction, we originally noted that the DW-EML has been applied in several of our group's publications using original data from both case–control pilot studies and validation studies with independent and prospective cohorts. In the revised manuscript, these self-citations have been removed from the main text and relocated to a dedicated supplementary section to minimize self-citation. We now explicitly state that while this approach has been extensively tested on original data by our group, the present study extends validation to independent datasets from other research groups, utilizing different instrumentation and analytic approaches than those previously reported. This further supports the robustness of the DW-EML methodology.

  • Despite the claim that permutation tests confirmed robustness, the very high accuracy scores (especially near 100%) suggest possible overfitting, particularly since metabolomic datasets are often noisy and variable.

Reply: High classification performance is not necessarily indicative of overfitting. While we acknowledge that the most reliable approach to assess overfitting involves validation with an independent dataset—and that such validation is not feasible in the cases presented in this study—our comprehensive data analysis does not indicate any signs of overfitting.

Both the permutation tests and cross-validation results consistently suggest that, to the best of our investigation, the models are not overfitted. Specifically, for the proposed ensemble model to produce misclassifications, a majority of the classifiers would need to make the same incorrect prediction. Unlike other ensemble strategies, our approach integrates diverse models trained using different underlying principles, making this scenario unlikely. Moreover, the weighting of individual model credibility and classification confidence further enhances the robustness and generalizability of our approach. We reported these considerations in the discussion part of the manuscript.

Additionally, our research group has successfully employed this ensemble scheme in two previous studies, where reliability was rigorously assessed through validation with independent cohorts comprising very large datasets. In both cases, the results confirmed the robustness and reliability of the approach. For further reference, please see: 10.3390/biom12091229 and 10.1016/j.ajog.2022.08.050.

  • The authors use multiple datasets from different sources but do not discuss potential batch effects, variations in sample collection, or differences in analytical methods.

Reply: We acknowledge the importance of considering potential batch effects, variations in sample collection, and differences in analytical methods when working with datasets from multiple sources. However, we would like to clarify that, although we utilized datasets from different sources, we did not merge or combine them. Instead, each dataset was analyzed independently, ensuring that potential inter-dataset variations did not influence the model training and validation process.

Moreover, all the datasets used in this study have been previously published in peer-reviewed journals, and the validity and reliability of the data have not been questioned in any prior evaluations. This further supports the robustness of the datasets employed in our analysis.

Finally, regarding batch effects, it is important to note that, in the presence of adequate randomization of samples across different batches, systematic batch effects would typically degrade global model performance. However, we did not observe such a decline in classification performance, suggesting that batch effects, if present, did not significantly impact our results.

Additionally, rather than being a limitation, the use of datasets from different sources and analytical methods represents an advantage of our work. A key objective of this study was to demonstrate that our model is applicable across different scenarios, making it a robust and generalizable approach. In contrast to our previous studies, where we applied the same ensemble scheme to multiple datasets collected within our laboratory under the same analytical conditions, this work serves as an independent validation, showing that the proposed framework maintains its effectiveness even with datasets obtained from diverse sources. This further highlights the flexibility and reliability of our approach. We added also this consideration in the paper discussion

  • The authors compare the proposed model with traditional PLS-DA but do not benchmark against other widely used ensemble learning techniques.

Reply: We appreciate the reviewer’s suggestion regarding benchmarking against other ensemble learning techniques. However, the comparison with PLS-DA was specifically conducted to evaluate the classification performance of the proposed DW-EML model relative to the results reported in previous studies that originally analyzed the same datasets. This allowed us to directly assess the improvement provided by our approach within the same experimental context.

Regarding the comparison with other ensemble models, it is important to note that the most widely used ensemble learning techniques in metabolomics primarily rely on a single classifier within bagging or boosting frameworks. These methods do not leverage the advantage of combining classifiers trained with diverse learning paradigms, as is the case in our ensemble approach. The key strength of DW-EML is precisely its ability to integrate models with different underlying learning logics, making it fundamentally different from traditional ensemble schemes used in this field.

Additionally, Random Forest (RF)—which is included as one of the classifiers within our DW-EML framework—can itself be considered an independent ensemble model, as it is inherently based on a bagging scheme of decision trees. This further supports the robustness of our approach, as it integrates both individual classifiers and a well-established ensemble method within a single, weighted framework.

  • It is unclear whether statistical tests (e.g., paired t-tests or Wilcoxon signed-rank tests) were conducted to confirm that the improvements are significant.

Reply:

  • The DW-EML model remains unclear on how metabolite importance is interpreted. Can the model outputs be linked to meaningful biological insights?

Reply: We appreciate the reviewer’s concern regarding the interpretability of metabolite importance in the DW-EML model. However, we would like to highlight that this aspect is explicitly addressed in our manuscript (lines 498-506):

"Indeed, the DW-EML approach also provides valuable insights into the relative importance of individual metabolites in the training of each classifier. By evaluating which metabolites carry the most weight in each model, researchers can develop a combined perspective that broadens the range of hypotheses that can be generated from untargeted metabolomic data using only PLS-DA. Metabolites selected consistently across different classifiers are likely to have significant biological relevance, and their consistent identification across multiple models strengthens confidence in their role. This level of assessment is only possible when multiple classifiers are combined, as the DW-EML approach does, making it a powerful tool for understanding complex metabolomic data and generating new biological insights."

The advantage of DW-EML is that it does not rely on a single model’s feature importance metric but rather integrates the weight assigned to each metabolite across different classifiers. This provides a more comprehensive and reliable assessment of metabolite importance, reducing biases that may arise from using a single modeling approach. By identifying metabolites that consistently contribute to classification across diverse learning paradigms, DW-EML strengthens the confidence in their biological relevance and helps generate new hypotheses in metabolomics research.

We believe this is a key strength of our approach and provides a valuable framework for extracting meaningful biological insights from complex metabolomic datasets.

Reviewer 2 Report

Comments and Suggestions for Authors

Review Report

Title: Double-Weighted Bayesian Model Combination for Metabolomics Data Prediction

The manuscript introduces a double-weighted Bayesian ensemble model for metabolomics data prediction. The study is relevant and timely, given the increasing use of machine learning in bioinformatics. However, the paper has major issues in grammar, structure, methodology justification, and citation consistency that needs a  Major Revisions. Below is a detailed review highlighting key areas for improvement.

  1. Language and Grammar Issues

The manuscript contains several grammatical errors, awkward phrasings, and unclear statements that reduce readability.

Incorrect:
"Double weighted Bayesian model combination ensemble machine learning model as a tool for description and prediction using metabolomics data."
 Corrected:
"Double-weighted Bayesian ensemble model for metabolomics data description and prediction." (More concise and grammatically correct.)

 Incorrect:
"A series of small simulation study are carried out."
 Corrected:
"A series of small simulation studies is carried out." (Incorrect verb agreement.)

 Incorrect:
"Ensuring a more reliable prediction framework."
 Corrected:
"Providing a more reliable prediction framework." (More natural phrasing.)

 Incorrect:
"An soft physiotherapy robot actuated by the soft pneumatic actuator is designed..."
 Corrected:
"A soft physiotherapy robot actuated by a soft pneumatic actuator is designed..." (Incorrect use of "an" before a consonant sound.)

 Recommendation:

  • Conduct thorough proofreading to improve grammar, readability, and sentence structure.
  1. Structural and Formatting Issues

Several structural problems disrupt the logical flow and clarity of the manuscript.

Identified Issues:

 Title Formatting Issues

  • Inconsistent capitalization.
     Suggested Title:
    "Double-Weighted Bayesian Model Combination for Metabolomics Data Prediction" (More professional and concise.)

 Table and Figure Formatting Issues

  • Tables (Table 1, 2, 3) have misaligned column headers.
  • Figures lack detailed captions explaining results.

 Equation Formatting Issues

  • Some mathematical equations appear without proper explanation or derivation.
  • Solution: Define each mathematical term before presenting an equation.

 Recommendation:

  • Ensure consistent formatting of titles, figures, and tables.
  • Add clear figure captions describing the significance of each result.
  1. Methodological Issues

The methodological framework needs further justification and more robust evaluation metrics.

  1. Justification of Model Selection

 Issue:

  • The manuscript does not compare the proposed Bayesian ensemble model with other ensemble methods, such as:
    • Random Forest
    • Gradient Boosting Machines (GBM)
    • Stacked Generalization

 Solution:

  • Provide a comparative justification for the selected ensemble method.
  • Compare performance with alternative models in a benchmark analysis.

  1. Performance Evaluation is Limited

 Issue:

  • The paper only reports accuracy, missing key evaluation metrics:
    • Precision
    • Recall
    • F1-score
    • ROC-AUC (if applicable)

 Solution:

  • Include additional classification metrics for a comprehensive evaluation.
  1. Robustness and Sensitivity Analysis

 Issue:

  • The paper mentions hyperparameter tuning (γ,σx,σr\gamma, \sigma_x, \sigma_r) but does not conduct sensitivity analysis.
  • What happens if hyperparameter values change?
  • Is the model robust across different settings?

 Solution:

  • Conduct sensitivity analysis by varying hyperparameters and reporting results.
  1. Citation and Reference Issues

The manuscript has multiple citation inconsistencies that need correction.

  1. Inconsistent Citation Style

 Issue:

  • Some citations use numbers ([1], [2]), while others use author names (Smith et al., 2020).
     Solution:
  • Ensure a consistent citation format (e.g., APA, IEEE, Chicago).
  1. Missing References for Key Claims

 Issue:

  • Some scientific claims about metabolomics applications in clinical settings lack citations.
     Solution:
  • Provide supporting references for all unverified claims.

Comments on the Quality of English Language

The manuscript demonstrates technical clarity but requires significant improvements in grammar, sentence structure, and consistency. Issues such as subject-verb agreement errors, redundancy, and passive voice overuse affect readability, and thorough proofreading or professional editing is recommended.

Author Response

The manuscript introduces a double-weighted Bayesian ensemble model for metabolomics data prediction. The study is relevant and timely, given the increasing use of machine learning in bioinformatics. However, the paper has major issues in grammar, structure, methodology justification, and citation consistency that needs a  Major Revisions. Below is a detailed review highlighting key areas for improvement.

  1. Language and Grammar Issues

The manuscript contains several grammatical errors, awkward phrasings, and unclear statements that reduce readability.

Incorrect:
"Double weighted Bayesian model combination ensemble machine learning model as a tool for description and prediction using metabolomics data."
 Corrected:
"Double-weighted Bayesian ensemble model for metabolomics data description and prediction." (More concise and grammatically correct.)

 Incorrect:
"A series of small simulation study are carried out."
 Corrected:
"A series of small simulation studies is carried out." (Incorrect verb agreement.)

 Incorrect:
"Ensuring a more reliable prediction framework."
 Corrected:
"Providing a more reliable prediction framework." (More natural phrasing.)

 Incorrect:
"An soft physiotherapy robot actuated by the soft pneumatic actuator is designed..."
 Corrected:
"A soft physiotherapy robot actuated by a soft pneumatic actuator is designed..." (Incorrect use of "an" before a consonant sound.)

 Recommendation:

  • Conduct thorough proofreading to improve grammar, readability, and sentence structure.

Reply: We appreciate the reviewer’s detailed feedback on language and grammar and fully acknowledge the importance of clarity and readability in scientific writing.

First, we accept the reviewer’s suggestion regarding the title change and will update it accordingly to improve conciseness and grammatical correctness.

Additionally, we sincerely apologize for the grammatical errors in the submitted version of the manuscript. Due to an oversight, an earlier draft was mistakenly sent for peer review before undergoing final language revision. However, we want to clarify that the manuscript has already been thoroughly revised by two native English-speaking co-authors to ensure proper grammar, sentence structure, and overall readability. The corrected version has now been carefully proofread, and all identified language issues have been addressed.

We appreciate the reviewer’s attention to detail and their constructive suggestions, which will help enhance the clarity and quality of the manuscript.

  1. Structural and Formatting Issues

Several structural problems disrupt the logical flow and clarity of the manuscript.

Identified Issues:

 Title Formatting Issues

  • Inconsistent capitalization.
     Suggested Title:
    "Double-Weighted Bayesian Model Combination for Metabolomics Data Prediction" (More professional and concise.)

Reply: We appreciate the reviewer’s suggestion regarding the title formatting and clarity. We have revised the title accordingly to "Double-Weighted Bayesian Model Combination for Metabolomics Data Prediction," as recommended.

Thank you for your valuable input in improving the professionalism and readability of the manuscript.

 Table and Figure Formatting Issues

  • Tables (Table 1, 2, 3) have misaligned column headers.
  • Figures lack detailed captions explaining results.

Reply: We appreciate the reviewer’s comments regarding the formatting of tables and figures.

Regarding the tables, we have ensured that they are formatted in accordance with the journal’s style guidelines. The alignment and structure comply with the required formatting standards. However, if there are specific misalignments that were caused during the typesetting or review process, we would be happy to make any necessary adjustments.

As for the figure captions, we have followed the editorial guidelines, which recommend writing captions that help describe and correctly interpret the figures rather than providing a summary of the results. Since the results are already discussed in detail within the main text, the captions are designed to complement the figures without unnecessary redundancy.

We appreciate the reviewer’s attention to detail and remain open to any further editorial suggestions.

 Equation Formatting Issues

  • Some mathematical equations appear without proper explanation or derivation.
  • Solution: Define each mathematical term before presenting an equation.

Reply: We appreciate the reviewer’s thorough assessment of the manuscript. However, we would like to clarify that our manuscript does not contain any mathematical equations. We have carefully reviewed the text to ensure clarity and confirm that no unexplained equations are present.

Additionally, we have verified that all acronyms used in the manuscript are properly defined in full at their first mention, ensuring clarity for the reader.

We appreciate the reviewer’s attention to detail and remain available for any further clarifications.

 Recommendation:

  • Ensure consistent formatting of titles, figures, and tables.
  • Add clear figure captions describing the significance of each result.
  1. Methodological Issues

The methodological framework needs further justification and more robust evaluation metrics.

  1. Justification of Model Selection

 Issue:

  • The manuscript does not compare the proposed Bayesian ensemble model with other ensemble methods, such as:
    • Random Forest
    • Gradient Boosting Machines (GBM)
    • Stacked Generalization

 Solution:

  • Provide a comparative justification for the selected ensemble method.
  • Compare performance with alternative models in a benchmark analysis.

Reply: We appreciate the reviewer’s suggestion regarding the justification and benchmarking of our DW-EML model. However, as reported in Tables 1, 2, and 3, both Random Forest and Gradient Boosted Trees were trained as individual models and subsequently integrated into the DW-EML framework. The results for these individual models serve as a benchmark analysis, allowing for a direct comparison between traditional ensemble methods and our proposed approach.

These results were already included in the original manuscript, which is why we did not add further comparisons. We believe this approach sufficiently justifies the selection of our ensemble method while also demonstrating its advantages over the individual classifiers.

We appreciate the reviewer’s attention to methodological rigor and remain open to any further clarifications.

  1. Performance Evaluation is Limited

 Issue:

  • The paper only reports accuracy, missing key evaluation metrics:
    • Precision
    • Recall
    • F1-score
    • ROC-AUC (if applicable)

 Solution:

  • Include additional classification metrics for a comprehensive evaluation.
  1. Robustness and Sensitivity Analysis

Reply: We appreciate the reviewer’s suggestion regarding the inclusion of additional evaluation metrics.

We would like to clarify that ROC-AUC values are already reported in all figures, providing a robust assessment of model performance. Additionally, precision, recall, and F1-score have been included in a dedicated Supplementary Material (S2), ensuring a comprehensive evaluation of the model’s classification performance.

 Issue:

  • The paper mentions hyperparameter tuning (γ,σx,σr\gamma, \sigma_x, \sigma_r) but does not conduct sensitivity analysis.
  • What happens if hyperparameter values change?
  • Is the model robust across different settings?

 Solution:

  • Conduct sensitivity analysis by varying hyperparameters and reporting results.

Reply: As detailed in Supplementary S2, we reported the performance of all models both with and without hyperparameter optimization. The results showed only minimal differences between the two conditions, highlighting the robustness of the models to this specific step.

  1. Citation and Reference Issues

The manuscript has multiple citation inconsistencies that need correction.

  1. Inconsistent Citation Style

 Issue:

  • Some citations use numbers ([1], [2]), while others use author names (Smith et al., 2020).
     Solution:
  • Ensure a consistent citation format (e.g., APA, IEEE, Chicago).

Reply: We appreciate the reviewer’s feedback regarding reference formatting.

All references have been reported following the journal’s editorial guidelines, using numerical citations as required. However, for the studies from which the data were collected, we opted to cite them using author names to enhance the readability of the manuscript and provide clear attribution to the original sources.

That said, we remain fully open to any editorial recommendations during the manuscript proofreading phase and will make any necessary adjustments to align with the journal’s final formatting requirements.

Missing References for Key Claims

 Issue:

  • Some scientific claims about metabolomics applications in clinical settings lack citations.
     Solution:
  • Provide supporting references for all unverified claims.

 Reply: We reported in the supplementary S1 a list of paper in which the reported DW-EML algorithm were used in clinical application using metabolomics data.

Reviewer 3 Report

Comments and Suggestions for Authors

I have two main concerns regarding this manuscript:

  1. The authors did not clearly state whether they developed the idea of the DW-EML model or if they simply applied an existing approach to the data. Ensemble models, including weighted ensemble methods, are not new, and it is important to clarify whether the novelty lies in the development of the DW-EML model or its application to metabolomics data.

  2. The authors used synthetic data to balance the imbalanced datasets. However, as some synthetic data generation methods can also denoise the data, it raises the question of how the authors confirmed the impact of the additional synthetic data on the final accuracy scores of the model. This point needs further clarification.

Minor points:

  1. In the introduction, the authors only mention PLS-DA and some other methods (Ref 9) without providing a broader overview of other commonly used methods in metabolomics. A more comprehensive discussion of alternative approaches would strengthen the introduction.

  2. The authors compared their results exclusively with other machine learning methods. Why did they not include comparisons with traditional methods like PLS-DA or other ensemble models? Including such comparisons would provide a more robust evaluation of the proposed method's performance.

Author Response

I have two main concerns regarding this manuscript:

  1. The authors did not clearly state whether they developed the idea of the DW-EML model or if they simply applied an existing approach to the data. Ensemble models, including weighted ensemble methods, are not new, and it is important to clarify whether the novelty lies in the development of the DW-EML model or its application to metabolomics data.

Reply: We appreciate the reviewer’s request for clarification regarding the novelty of our approach.

To the best of our knowledge, the proposed DW-EML framework, with these specific two weighting schemes, has never been previously reported. Our research group has successfully applied this system for several years using original metabolomics data generated in our laboratory (please refer to Supplementary S1 for further details).

In this specific study, we aimed to extend and validate our approach by applying it to publicly available datasets obtained through different analytical techniques. This represents a significant step forward in demonstrating the generalizability and robustness of the DW-EML model in metabolomics research.

We appreciate the opportunity to clarify this point and highlight the novelty of our work.

  1. The authors used synthetic data to balance the imbalanced datasets. However, as some synthetic data generation methods can also denoise the data, it raises the question of how the authors confirmed the impact of the additional synthetic data on the final accuracy scores of the model. This point needs further clarification.

Reply: We acknowledge this consideration as a study limitation. Nevertheless, it is important to emphasize that managing class imbalance is essential when training a machine learning system. In metabolomics, the inherent scarcity of samples relative to the number of features renders undersampling strategies impractical, thus necessitating alternative approaches.

Minor points:

  1. In the introduction, the authors only mention PLS-DA and some other methods (Ref 9) without providing a broader overview of other commonly used methods in metabolomics. A more comprehensive discussion of alternative approaches would strengthen the introduction.

Reply: We appreciate the reviewer’s suggestion to provide a broader overview of classification methods in metabolomics. However, as we have already discussed in the manuscript, while any classification model can, in principle, be applied to metabolomics studies, the reality is that PLS-DA remains the overwhelmingly predominant method due to both tradition and practical convenience.

In this context, listing other classification models would be somewhat instrumental, as their actual application in metabolomics remains extremely limited. To further support this point, we have now added a specific reference that discusses the widespread use of PLS-DA in metabolomics research: https://www.sciencedirect.com/science/article/abs/pii/S0003267015001889.

We appreciate the reviewer’s attention to strengthening the introduction and believe this addition provides further clarity on the methodological landscape in metabolomics.

  1. The authors compared their results exclusively with other machine learning methods. Why did they not include comparisons with traditional methods like PLS-DA or other ensemble models? Including such comparisons would provide a more robust evaluation of the proposed method's performance.

Reply: We appreciate the reviewer’s suggestion regarding the inclusion of comparisons with traditional methods. However, it is unclear what the reviewer means by distinguishing PLS-DA from machine learning methods, as PLS-DA itself is a machine learning technique.

In our study, we explicitly compared the results of DW-EML with those obtained using PLS-DA, referring to the classification performance reported in the original studies that analyzed the same datasets. This allowed us to directly assess the advantages of our approach relative to a widely used method in metabolomics.

Furthermore, our DW-EML framework incorporates two well-established ensemble models: Random Forest (RF) and Gradient Boosted Trees (GBT). These models serve as benchmarks for other ensemble learning methods, demonstrating the added value of integrating multiple classifiers with diverse learning paradigms.

Given that our study already includes comparisons with both PLS-DA and two ensemble models, we believe our evaluation is sufficiently comprehensive. However, we remain open to any specific suggestions the reviewer may have regarding additional benchmarks.

Round 2

Reviewer 1 Report

Comments and Suggestions for Authors

I recommend it for publication.

Comments on the Quality of English Language

Good

Author Response

Thanks

Reviewer 2 Report

Comments and Suggestions for Authors

The revised version can be accepted for publication.

Author Response

Thanks

Reviewer 3 Report

Comments and Suggestions for Authors

Thank you for your response and for acknowledging the limitation regarding the use of synthetic data. While I understand the necessity of addressing class imbalance in metabolomics, particularly given the challenges posed by the scarcity of samples relative to the number of features, my concern remains regarding the potential impact of synthetic data on the model's performance. Specifically, synthetic data generation methods can sometimes inadvertently denoise or alter the data distribution, which may artificially influence the final accuracy scores.

It would be helpful if you could elaborate further on any steps taken to assess the impact of the synthetic data on model performance. For example, were any ablation studies, comparisons with alternative methods, or validations on independent datasets conducted to ensure that the observed improvements were not solely due to the synthetic data generation process? Addressing this point would strengthen the robustness of your findings.

Author Response

Thank you very much for your insightful comment and for highlighting an important aspect of our study. We acknowledge your concern regarding the potential unintended effects of synthetic data generation on model performance, particularly considering possible data denoising or alterations in the original feature distribution.

To address your points, we performed an ablation study specifically on a dataset with reduced class imbalance (the CB dataset from Arturas Grauslys et al.). In this scenario, we observed that the synthetic data augmentation method contributed positively to model performance without introducing significant distortion of the original data distribution. Conversely, attempts using undersampling techniques consistently resulted in overfitting across multiple datasets, indicating that synthetic data augmentation provided a more robust approach. We have detailed these results comprehensively in a dedicated section within the supplementary materials.

Regarding validation on independent external datasets, such validation was not feasible within the current study due to our experimental design, which intentionally employed publicly available datasets. However, as previously highlighted in both the Introduction and Discussion sections, our research group has successfully validated this methodological approach using prospective and independent datasets in other published studies.

We will update the manuscript to clearly emphasize these points, ensuring transparency about the limitations and advantages associated with synthetic data augmentation.